# Novel Transcriptional and DNA Methylation Abnormalities of SORT1 Gene in Non-Small Cell Lung Cancer

**DOI:** 10.3390/cancers16112154

**Published:** 2024-06-06

**Authors:** Amelia Acha-Sagredo, Cornelia M. Wilson, Naiara Garcia Bediaga, Helen Kalirai, Michael P. A. Davies, Sarah E. Coupland, John K. Field, Triantafillos Liloglou

**Affiliations:** 1Institute of Systems, Molecular & Integrative Biology, University of Liverpool, Liverpool L69 3BX, UK; amelia.acha@crick.ac.uk (A.A.-S.); mpdavies@liverpool.ac.uk (M.P.A.D.); j.k.field@liverpool.ac.uk (J.K.F.); 2Life Sciences Industry Liaison Lab, School of Psychology and Life Sciences, Canterbury Christ Church University, Canterbury CT1 1QU, UK; cornelia.wilson@canterbury.ac.uk; 3Adelaide Centre for Epigenetics, Faculty of Health and Medical Sciences, The University of Adelaide, Adelaide, SA 5005, Australia; naiara.garciabediaga@osakidetza.eus; 4Institute of Life Course and Medical Sciences, University of Liverpool, Liverpool L69 3BX, UK; hkalirai@liverpool.ac.uk (H.K.); amelie@liverpool.ac.uk (S.E.C.); 5Medical School, Edge Hill University, St Helens Road, Ormskirk L39 4QP, UK

**Keywords:** sortilin, non-small cell lung cancer, DNA methylation, splicing

## Abstract

**Simple Summary:**

Sortilin is a protein with tumour-suppressor activity. Among other functions, sortilin moves around other proteins in the cell and also out of it by leading them to small vesicles termed exosomes. The aim of this study was to establish the degree of abnormal expression of the gene in non-small cell lung cancer. We show here that the alternative forms of the gene are affected both qualitatively and quantitatively. In addition, the ratio of their expression changes. We demonstrate that expression changes of the SORT1 form are due to DNA methylation of its regulatory region. As sortilin is involved in trafficking EGFR, a known target of current drugs, it is possible that these alterations, beyond their involvement in cancer development, may be predictive of the response to these therapies.

**Abstract:**

Sortilin is an important regulator with potential tumour-suppressor function by limiting EGFR signalling. In this study, we undertook a comprehensive expression analysis of sortilin transcript variants and the DNA methylation status of their corresponding promoters in human non-small cell carcinomas (NSCLCs). RNA/DNA was extracted from 81 NSCLC samples and paired normal tissue. mRNA expression was measured by qPCR and DNA methylation determined by pyrosequencing. BigDye-terminator sequencing was used to confirm exon-8 alternative splicing. Results demonstrated that both SORT1A and SORT1B variants were downregulated in lung tumours. The SORT1A/SORT1B expression ratio was higher in tumours compared to normal tissue. SORT1B promoter hypermethylation was detected in lung tumours compared to normal lung (median difference 14%, Mann–Whitney test *p* = 10^−6^). Interestingly, SORT1B is hypermethylated in white blood cells, but a small and very consistent drop in methylation (6%, *p* = 10^−15^) was observed in the lung cancer cases compared to control subjects. We demonstrate that the SORT1B exon-8 splice variation, reported in sequence databases, is also a feature of SORT1A. The significantly altered quantitative and qualitative characteristics of sortilin mRNA in NSCLC indicate a significant involvement in tumour pathogenesis and may have significant impact for its utility as a predictive marker in lung cancer management.

## 1. Introduction

Lung cancer is a complex, largely smoking-related disease involving multiple genetic and epigenetic alterations, and it is the most common cause of cancer death worldwide (1.76 million) [1]. Approximately 80% of all primary lung cancer diagnoses are non-small cell lung carcinoma (NSCLC), with the most common subtypes within this group being squamous cell carcinoma (SCC), adenocarcinoma, and large-cell carcinoma [2]. The 5-year survival rate depends on the disease stage according to the TNM system [3]; therefore, beyond primary prevention (e.g., smoking secession), reduction in mortality can be achieved by earlier diagnosis, e.g., through implementing low-dose CT screening [4]. A wide spectrum of genetic and epigenetic alterations is the driving force of lung carcinogenesis [5], including mutations, copy-number variations, splicing deregulation, DNA methylation, non-coding RNAs, and histone modifications. Some molecular abnormalities established in NSCLC are important determinants for clinical stratification for targeted therapies. However, the major challenge is heterogeneity and the multiplicity of those molecular abnormalities, which essentially create very diverse clinical phenotypes. Identification of common abnormalities provides additional opportunities for new treatments, improved stratification, or earlier diagnosis.

Sortilin is a family member of the vacuolar protein sorting 10 (VPS10)-domain receptors and is a multifaceted receptor of neurotrophin factors as well as a co-receptor of tyrosine receptor kinases, cytokine receptors, G protein-coupled receptors, and ion channels [6]. The VPS10 domain constitutes the whole of the sortilin extracellular luminal region, which displays two structural features: an N-terminal portion forming a β-propeller and a C-terminal portion containing 10 conserved cysteines, which are crucial for ligand interaction and key for the function of sortilin [7]. Sortilin is an important regulator of neuronal viability, development, plasticity, and function [8,9]. Notably, sortilin is expressed in the central nervous system (CNS) but can also be found in other tissues and organs [10]. Its function outside the CNS remains largely elusive, but emerging evidence implicates that deregulation of sortilin could be a contributing factor in human diseases [11,12].

Human sortilin is encoded by the SORT1 gene (1p13.3) [13], which produces two transcript variants (SORT1A and SORT1B) utilising alternative promoters and transcription start sites. SORT1A represents the longer transcript and encodes the longer isoform (831 aa), while SORT1B has a different 5′ UTR and coding sequence that results in a shorter isoform at the N-terminus (694 aa). Furthermore, based on the available transcript sequences in NCBI (Gene ID: 6272), compared to SORT1A, SORT1B lacks an internal amino acid (278K/279A −> 278T) due to the use of an alternative splice junction at the 5′ end of exon 8.

The status of sortilin expression in human cancer, based on the existing literature, is currently unclear, although there is evidence to support an oncogenic role for it in gastrointestinal cancers [14,15,16] as well as thyroid [17], cervical [18], and lung carcinomas [19]. In addition, in vitro evidence in glioblastoma [20] and breast cancer cells [21] supports such a role. However, it has been shown that sortilin can limit EGFR signalling by promoting its internalisation [22], demonstrating a potential tumour-suppression function in the context of cancers dependent on receptor kinase signalling pathways.

In the absence of comprehensive evidence of the status of sortilin in NSCLC, the aim of this study was to extensively explore the expression and DNA methylation levels of SORT1 variants in both primary NSCLC samples and normal lung tissue as well as peripheral blood and cell lines of lung cancer origin.

## 2. Materials and Methods

### 2.1. Patients and Samples

A set of 81 NSCLC surgical samples from chemotherapy/radiotherapy-naïve patients from the Liverpool Lung project (LLP) biobank was used for SORT1 expression studies. Frozen tissues were macrodissected to ensure >70% tumour content. For DNA methylation analysis, we utilised DNAs from (a) 123 tumour and 37 paired adjacent normal, treatment-naïve, surgical samples available from previous studies [23] and (b) peripheral blood from 254 pre-treatment lung cancer cases and 248 age/sex-matched control individuals from the LLP biobank. The latter are recruits from the population cohort element of the LLP study that had confirmed absence of lung cancer for at least 5 years after the blood sample was taken. Clinicopathological characteristics of the patients used for expression and methylation studies are provided in Appendix A; briefly, most were male (68% expression; 74% methylation), SCC (69% expression; 57% methylation), T1 or T2 (85% expression; 84% methylation), node-negative (52% expression; 51% methylation), and moderately differentiated (53% expression; 62% methylation). Demographic and clinical characteristics of the population utilised for DNA methylation analysis in peripheral blood are summarised in Appendix A. All patients were recruited from Liverpool Heart and Chest Hospital under appropriate ethical approval, and informed and voluntary consent was obtained.

### 2.2. Cell Lines and Growth Conditions

SORT1 status was examined in NSCLC cell lines to determine if this is an important characteristic for the retention of a malignant phenotype in culture environment. Thirteen NSCLC cell lines (A549, Calu-1, Calu-3, Calu-6, COR-L23, DMS53, H358, H2073, HTB-59, HTB-182, LUDLU-1, SK-LU-1, and SK-MES-1) and a non-tumourigenic human foetal lung fibroblast cell line (IMR-90) were cultured in Dulbecco’s Modified Eagle’s Medium (DMEM)/Ham’s Nutrient Mixture F-12 (1:1) containing 10% foetal bovine serum (Sigma-Aldrich, Gillingham, UK) at 37 °C and 5% CO_2_. All the cells were mycoplasma-free when tested using the e-Myco™ plus Mycoplasma PCR Detection Kit (iNtRON Biotechnology, Seongnam-si, Republic of Korea) and were authenticated using the GenePrint 10 System (Promega, Southampton, UK) on a 3130 Genetic Analyzer (Life Technologies, Paisley, UK).

### 2.3. RNA Isolation

Ten tissue sections of 10 μm thickness were used for RNA extraction from each sample. Total RNA was extracted from both tissue and cell lines using the ZymoResearch Direct-zol mini kit (Cambridge Bioscience, Cambridge, UK). RNA concentration and purity were determined by OD at 260/280/230 nm using a NanoDrop ND-1000 spectrophotometer (Thermo Fisher Scientific, Hemel Hempstead, UK).

### 2.4. SORT1A and SORT1B Quantitative RT-PCR

Quantitative RT-PCR (qRT-PCR) assays utilising hydrolysis probes were designed to analyse the expression levels of SORT1A and SORT1B (Appendix A). First, 1 µg of total RNA was reverse-transcribed using the QuantiTect Reverse Transcription Kit (Qiagen, Manchester, UK) according to the manufacturer’s instructions. Quantitative real-time PCR was performed in triplicate in a final reaction volume of 15 μL (containing 7.5 μL QuantiTect probe PCR mastermix (Qiagen, UK), 900 nM of each primer, 250 nM of probe, and 3 μL of cDNA) on an Applied Biosystems 7500 real-time PCR machine under the following thermal profile: 95 °C for 15 min, 45 cycles of 94 °C for 15 s, 58 °C for 30 s, and 60 °C for 45 s). The expression of each SORT1 variant was quantified relative to TBP expression using the 2^−ΔΔCt^ method [24].

### 2.5. DNA Extraction and Methylation Analysis

DNA was isolated using the DNeasy blood and tissue kit (Qiagen, UK) and quantified using a NanoDrop ND-1000 spectrophotometer (Thermo Fisher Scientific, Hemel Hempstead, UK). First, 1 µg of DNA was bisulphite-converted using the EZ-96 DNA Methylation-Gold Kit (ZymoResearch, Irvine, CA, USA). DNA methylation analysis was performed by pyrosequencing (Qiagen, UK) as previously described. Primers and PCR thermal profiles for each pyrosequencing assay are provided in Appendix A. Hyper/hypo-methylation thresholds were calculated using the 95% reference interval (mean ± 2 × SD) of the normal sample values.

### 2.6. SORT1A Exon-8 Sequencing

Amplification and sequencing primers (Appendix A) were designed to specifically amplify exon 8 on SORT1A. PCR reactions were performed using HotStarTaq Plus Mastermix (Qiagen, UK), 150 nM of each primer, and 3 µL of cDNA. The thermal profile was 95 °C for 5 min, 40 cycles of 94 °C for 30 s, 59 °C for 30 s, and 72 °C for 60 s, with a final extension step at 72 °C for 10 min. Amplicon size was confirmed by 2% agarose gel electrophoresis, and sequencing reactions were carried out after PCR products were cleaned up using the QIAquick PCR Purification Kit (Qiagen, UK). Sequencing reactions were performed with the sequencing primer (Appendix A) using the BigDye Terminator v3.1 Cycle Sequencing Kit (Life Technologies, Paisley, UK) on an ABI 3130 Genetic Analyzer (Life Technologies, Paisley, UK).

### 2.7. SORT1A Exon-8 Fragment Analysis

Capillary electrophoresis-based fragment analysis was carried out to assess the ratio of SORT1A exon-8 alternatively spliced transcripts (SORT1A1 and SORT1A2) with specific FAM-labelled reverse and unlabelled forward primers (Appendix A). PCR amplifications were performed using HotStarTaq Mastermix (Qiagen, UK), 170 nM primers, and 3 μL cDNA under the following conditions: 95 °C for 15 min followed by 38 cycles of 94 °C for 30 s, 62 °C for 30 s, and 72 °C for 45 s, with a final extension step at 72 °C for 20 min. Then, 1 µL of PCR product was mixed with loading solution (size marker ROX-1000 and formamide in a ratio 1:19), and after denaturation at 95 °C for 5 min, samples were analysed on a 3130 Genetic Analyzer (Life Technologies, Paisley, UK). The ratio of spliced transcripts was calculated by dividing the peak area for the longer SORT1A1 (+AAG) by the shorter SORT1A2 (−AAG) spliced transcripts.

### 2.8. Statistical Analysis

Non-parametric tests were used for statistical analyses (PASW Statistics 24.0, SPSS), as one-sample Kolmogorov–Smirnov test showed skewed distribution of continuous variables. Pairwise comparisons between normal and NSCLC samples were performed with Wilcoxon test. Differences in frequencies among nominal variables were determined with Pearson’s chi-square test. Mann–Whitney tests were used to assess the differences for continuous variables among independent groups within clinical parameters. Bivariate correlation was assessed with Spearman’s test. Patients’ survival differences were calculated with the log-rank test.

## 3. Results

### 3.1. SORT1 Expression Levels in Human NSCLC Cancer Tissue

We determined the mRNA expression of both SORT1 transcript variants (SORT1A and SORT1B) by RT-qPCR in 81 paired lung cancer and histologically normal adjacent tissues. SORT1A was significantly downregulated (Mann–Whitney test, *p* = 10^−13^) in lung cancer tissues compared to paired adjacent normal lung tissues (Figure 1A). Likewise, the expression of SORT1B was also reduced in NSCLC samples compared to paired non-tumour lung tissue (*p* = 10^−14^; Figure 1B). SORT1A and SORT1B transcript levels correlated in normal adjacent tissue (ρ = 0.688, *p* = 10^−12^; Figure 1C), but this correlation was weaker in lung tumour tissue (ρ = 0.476, *p* = 10^−5^; Figure 1D). The SORT1A/SORT1B expression ratio was found to be higher in tumours compared to paired normal tissue (*p* = 10^−12^; Figure 1E).

To validate our findings, we interrogated expression data in the public domain utilising the GEPIA tool [25]. It was evident in these data that the SORT1 mRNA expression is highly variable among different human cancers (Appendix A), with both lung adenocarcinomas and SCC displaying loss of expression, while cancers such as liver, ovarian, and pancreatic carcinomas demonstrate overexpression. It must be noted that these data report total SORT1 mRNA expression.

The expression profile of SORT1A and SORT1B was also assessed in a panel of 13 lung cancer cell lines along with a normal fibroblast lung cell line (IMR90). SORT1A expression was significantly reduced in all lung cancer cell lines (Appendix A). Interestingly, the expression of SORT1B was completely absent or markedly reduced in lung cancer cell lines (Appendix A).

Upon assessing the potential associations of SORT1A and SORT1B with the patients’ clinicopathological parameters, only a borderline association (Mann–Whitney test, *p* = 0.046) of higher SORT1A expression (i.e., reduced loss of expression) in adenocarcinomas compared to SCC was seen. No other significant association between SORT1A or SORT1B levels and any other clinicopathological parameters was observed. Due to the limited availability of survival information in our dataset, we examined patient survival in relationship to SORT1 expression in publicly available datasets using the KMplotter (Figure 2) and Xena (Appendix A) tools [26,27]. Interestingly, in all the datasets analysed, a low SORT1 expression was associated with poor overall and disease-free survival in adenocarcinomas, while in SCC, either non-significant or borderline trends with favourable outcome were derived between SORT1 expression and survival.

### 3.2. Promoter Methylation of SORT1 Transcription Variants in Human NSCLC Cancer Tissue

We investigated whether the observed altered expression of SORT1 transcript variants in NSCLC was associated with changes in DNA methylation since the alternative transcripts are driven by their individual promoters. Pyrosequencing methylation analysis assays were designed to cover multiple CpGs in the promoter and 5′-UTR regions of SORT1A and SORT1B genes to allow for quantitation of their DNA methylation levels in NSCLC samples and cell lines.

Initially, the methylation status of both variants was assayed in 37 NSCLC and paired normal adjacent tissue. SORT1A promoter was unmethylated in all the normal and tumour samples tested (Figure 3A). In contrast, considerable but variable methylation levels were observed for SORT1B. Based on these initial results, we decided to analyse the methylation levels of SORT1B in 84 additional tumour samples. Cumulative analysis of all the samples (37 normal and 121 tumours) demonstrated significantly higher levels of methylation in lung tumours (median methylation = 35.5%) compared to adjacent normal tissues (median methylation = 21.5%, Mann–Whitney test *p* = 10^−6^; Figure 3B). SORT1B hypermethylation (defined as the 95% reference range of normal tissue readings) was observed in 32/69 (46.4%) SCC compared to only 7/51 (13.7%) in adenocarcinomas (chi-square test, *p* = 0.00016). No further associations were observed between SORT1B methylation and any other clinicopathological parameters. When assessing the association between the expression and the methylation status of SORT1B in 41 lung tumours with both RNA and DNA availability from the same specimen, no direct correlation was found (Spearman’s correlation, rho = 0.067, *p* = 0.679). No associations between survival and SORT1B promoter methylation could be attempted, as KMplotter and Xena platforms do not hold NDA methylation information.

We also determined the methylation status of SORT1A and SORT1B in NSCLC cell lines. SORT1A showed, similar to the primary NSCLC tissue, low or zero levels of methylation (Figure 3C). In contrast, cancer cell lines exhibited varying levels of SORT1B methylation (Figure 3D).

### 3.3. Promoter Methylation of SORT1B in Peripheral Blood from NSCLC Patients

The methylation status of SORT1B promoter was assessed in peripheral blood mononuclear cells (PBMC) from 254 lung cancer patients and 248 age- and sex-matched controls. A small (~6%), but very consistent difference was observed in SORT1B promoter methylation levels in PBMCs from lung patients compared to controls (median methylation: 39.1% vs. 45.2%, Mann–Whitney test *p* = 10^−15^; Figure 4A). Low SORT1B methylation was statistically significant for each individual histological classification (adenocarcinomas, squamous carcinomas, and small cell carcinomas) (Figure 4C). No significant associations were found between the levels of SORT1B methylation in blood and clinicopathological parameters such as patients’ sex, tumour histology, stage, differentiation, and nodal involvement. The DNA methylation of the retrotransposable element LINE-1 was assessed as a measure of global DNA methylation level [28], demonstrating no differences in the blood cells of lung cancer cases compared to controls (Figure 4B,D), suggesting that the reduction observed in SORT1B promoter methylation is not a passive consequence of global DNA demethylation.

### 3.4. SORT1A Alternative Splice Variants

According to the NCBI Gene database, the SORT1B variant utilises an alternate in-frame splice junction at the 5′ end of exon 8 that leads to the loss of an internal amino acid compared to SORT1A. We tested if this alternative splice site was also used in SORT1A. We selected six pairs of NSCLC samples and normal adjacent tissue and specifically amplified and sequenced the 5′ end of exon 8 on SORT1A transcripts. Sequencing of both NSCLC samples and adjacent normal tissues produced clean electropherograms up to the beginning of exon 8, where a mixed (3 bp insertion) signal was detected (Appendix A), demonstrating that there was a mixed population of SORT1A transcript splice variants in all specimens.

We expanded our initial observation on the balance between SORT1A in-frame alternative splice variants in 48 paired lung cancer and histologically normal adjacent tissues by capillary-based fragment analysis. We observed that the transcript carrying the AAG triplet (SORT1A1) was the most abundant in both tumour and normal adjacent tissues. However, the ratio of SORT1A1:SORT1A2 variants was significantly lower in lung cancer compared to paired adjacent normal lung tissue (median ratio: 1.71 vs. 2.14, Wilcoxon test *p* = 0.002; Figure 5A). In tumours, no significant associations were found between SORT1A splice variant ratio and histology, differentiation, or lymph node involvement. However, a borderline association was observed with sex, demonstrating a lower SORT1A1:SORT1A2 ratio in females compared to males (1.64 vs. 1.99, Mann–Whitney test *p* = 0.033; Figure 5B).

The functional significance of these splice variants is not yet understood. The predicted protein structure of the two variants was created using the Expasy tool [29]. The predicted model demonstrates a potentially important structural difference between the two variants (Appendix A) at residues 278–279, which are in the VPS10 domain of the protein.

## 4. Discussion

Sortilin emerges as an important regulator for normal cell physiology, potentially affecting a wide spectrum of diseases, including cancer [11]. There is already preliminary evidence to suggest that sortilin expression in tissue or the soluble form of the protein in plasma may have important utility for the management of cancer [14,20].

The interaction of sortilin with Trk receptors and p75NTR stimulates PI3K, MAPK, and PLC-g pathways that are involved in cell survival and differentiation [11], while the apoptotic pathway is activated through interactions of proneurotrophins with sortilin and p75NTR complex. Sortilin interacts with several tyrosine kinase receptors such as EGFR and TrkB, affecting the MAPK pathway, PI3K/AKT/mTOR, and JAK/STAT pathways that control cell proliferation, migration, invasion, anti-apoptosis, and pro-angiogenesis [22] Cross-talk exists between NF-kB pathway and the apoptosis pathway through the binding of neurotrophins to p75NTR [11]. Given the diverse role of sortilin, epigenetic changes in SORT1 could alter sortilin’s specificity to interact with different co-receptors and thus impact the signalling pathways associated with these receptors.

In this study, we aimed to clarify the status of SORT1 mRNA expression in NSCLC, following our pilot observations, which demonstrated downregulation of SORT1 in this tumour type, contrasting the published data in other human cancers. Using primary NSCLC surgical specimens, we verified the downregulation of SORT1 in both adenocarcinomas and SCC. This result is in complete concordance with public domain data, which has also confirmed that the SORT1 gene is upregulated or downregulated in different cancer types. The mechanisms behind these contradictory associations are currently unknown and are most probably related to epigenetic programming underlying tissue differentiation as well as the causal factors of each cancer type. Nevertheless, the SORT1 gene appears to potentially act as an oncogene or as a tumour suppressor gene in different cancer types. In fact, a highly intriguing finding from our study is that while SORT1 is downregulated in both adenocarcinoma and SCCs, low SORT1 expression is associated with poor outcome in adenocarcinomas, while it marginally appears to be favourable for patients with squamous cancers of the lung. Whilst the regulation of EGFR activity by sortilin [22] seems to be the most plausible explanation behind this, given that EGFR signalling in the lung has more important clinical significance for adenocarcinomas, it can be considered highly probable that more pathways are involved.

A unique feature of this study is the thorough investigation of the different transcripts of the SORT1 gene in NSCLC. There are two alternative transcripts (SORT1A and SORT1B) driven by different promoters. Our analysis demonstrated that the expression ratio SORT1A/SORT1B significantly increases in NSCLC. While the functional significance of SORT1B is currently not understood, it is predicted to encode a shorter protein variant. However, to date, there is no convincing evidence in the literature to show this specific variant is expressed at the protein level. In our Western blot analysis in cell lines with detectable SORT1B transcripts, we could not detect the SORT1B protein, even though the antibody used is raised against a common SORT1A–SORT1B epitope. Among the current hypotheses behind this absence is that SORT1B may be encoding a soluble form of sortilin [30] or that it may be a long non-coding transcript that regulates the SORT1A half-life and/or translation rate; however, these have to be experimentally proven in future work.

We also demonstrated in this study the presence of a splicing variation at the beginning of exon 8 for SORT1A, which until now was considered a distinct feature of SORT1B. The combination of transcriptional start site and splicing variation therefore gives rise to four alternative transcripts (SORT1A1, SORT1A2, SORT1B1, and SORT1B2). The functional differences of these transcripts are still to be elucidated, but they could play distinct cellular roles in response to diverse external stimuli.

In addition, the splicing variant ratio of SORT1A1/SORT1A2 changed significantly in NSCLC tissue compared to normal lung tissue, indicating a potential functional role. In the absence of functional evidence, protein structure prediction is in favour of such a difference, but this requires experimental confirmation. Equally, the functional difference of SORT1A and SORT1B transcript variants is unclear and requires further research. It is of interest that in most of the established NSCLC cell lines used in this study, SORT1B expression was reduced beyond detectable levels, suggesting that the disruption of the SORT1A/SORT1B balance may be important for cell survival. This is also supported by our finding that the SORT1A/SORT1B ratio is higher in primary NSCLC tumours compared to normal adjacent lung tissue.

This study shows that SORT1B promoter is highly hypermethylated in many NSCLC samples compared to normal lung tissue, while SORT1A promoter is not subject to such regulation. One of the most intriguing findings of this study, however, is the low but very consistent difference in the DNA methylation of the SORT1B promoter in the peripheral blood of NSCLC and age/sex-matched controls. Our findings demonstrate a 6% lower SORT1B promoter methylation in the white blood cells of NSCLC cases compared to controls. Although highly statistically significant, the low amplitude of the difference denotes that this is a feature of a subpopulation of white cells. While a future study in distinct white cell subpopulations may identify those carrying this epigenetic difference, it is very likely that this reflects an immune response to the tumour or a related coexisting inflammatory condition.

The evidence in this study clarifies that the SORT1 gene is downregulated in NSCLC and demonstrates novel transcriptional and epigenetic irregularities. As this is an observational study, it does not provide functional relationships. Our findings provide solid support for the involvement of abnormal SORT1 regulation in lung cancer pathogenesis and enables the generation of strong hypotheses, opening up multiple avenues for future functional analysis of the gene. These results also pose a challenge for screening SORT1 irregularities in other cancer types, given that differential expression and associations with cancer survival vary considerably. Overall, our data add significant support to the existing hypothesis that sortilin is an important regulator of cancer pathogenesis and that it may be utilised as a biomarker or potentially as a therapeutic target for cancer management.

## Figures and Tables

**Figure 1 cancers-16-02154-f001:**
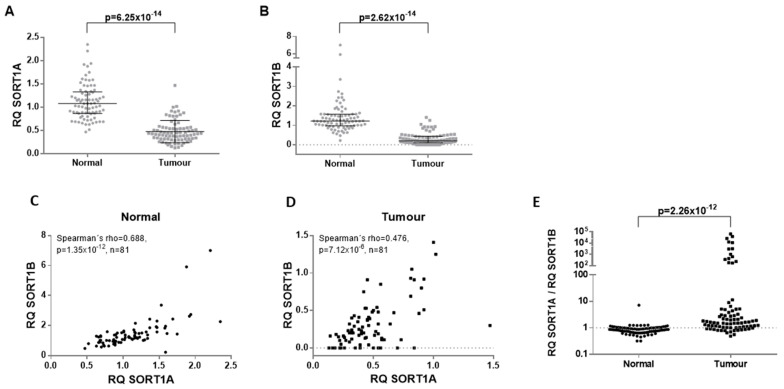
mRNA expression of SORT1A and SORT1B transcripts in primary NSCLC and paired adjacent normal lung tissue. There is an apparent downregulation of both SORT1A (**A**) and SORT1B (**B**) transcripts. Interestingly, the correlation coefficient between SORT1A and SORT1B transcript levels in normal lung (**C**) is higher than the one observed in NSCLC tissue (**D**), indicating a potential distortion of SORT1A/1B transcription in cancers. The comparison of SORT1A/SORT1B mRNA expression ratio in primary NSCLC and paired adjacent normal lung tissue demonstrates a higher SORT1A/SORT1B ratio in cancerous tissue (**E**).

**Figure 2 cancers-16-02154-f002:**
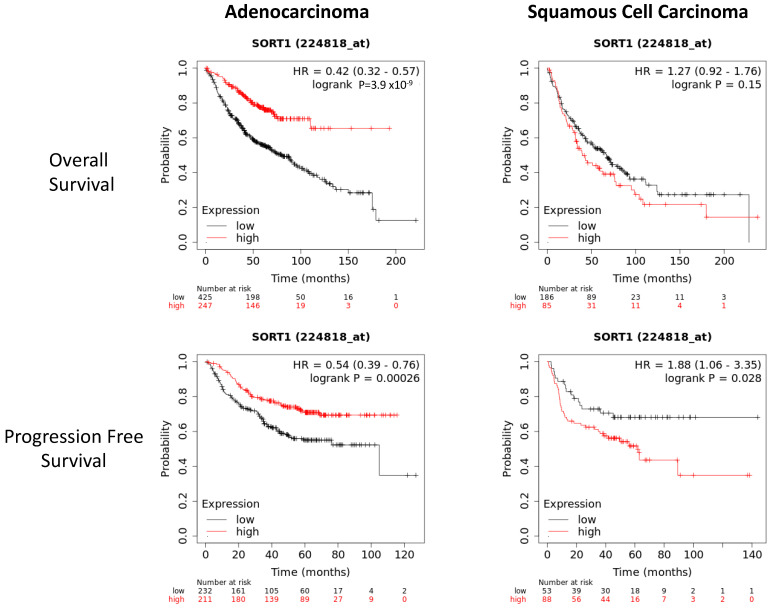
Assessment of the survival according to SORT1 expression in patients with NSCLC. The survival plot data from KMplotter tool indicates that low SORT1 expression is associated with poor overall and disease-free survival in lung adenocarcinomas. Conversely, there is a borderline trend of low SORT1 expression in lung squamous cell carcinomas with favourable outcome.

**Figure 3 cancers-16-02154-f003:**
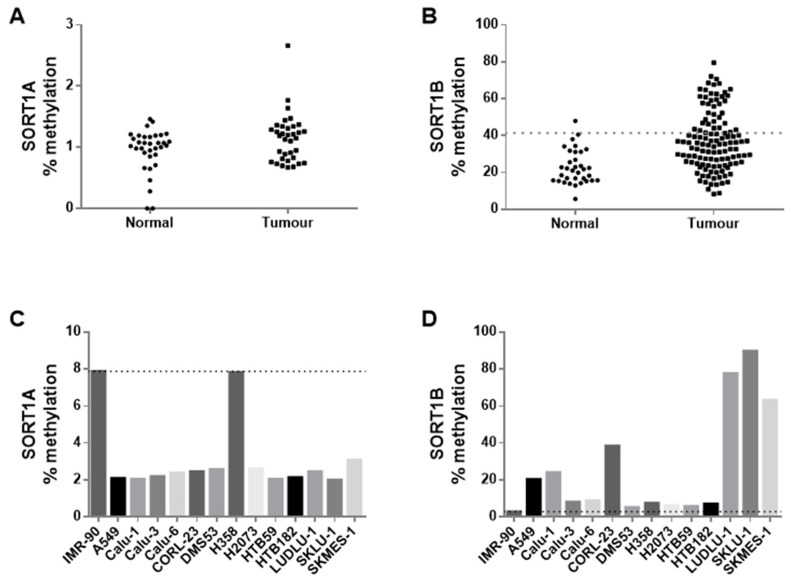
DNA methylation analysis of SORT1A and SORT1B promoters. SORT1A is unmethylated (<3% methylation detected) in both lung normal and tumour specimens (**A**) and in most cell lines (**C**). In contrast, higher SORT1B DNA methylation levels are detected in normal lung tissue with a significant hypermethylation observed in most lung tumours (**B**). Similarly, high DNA methylation levels are detected in 6/13 lung cancer cell lines (**D**). The dotted line represents the threshold above which a sample is considered to be hypermethylated (95% reference interval (mean + 2 × standard deviation) of the normal sample values).

**Figure 4 cancers-16-02154-f004:**
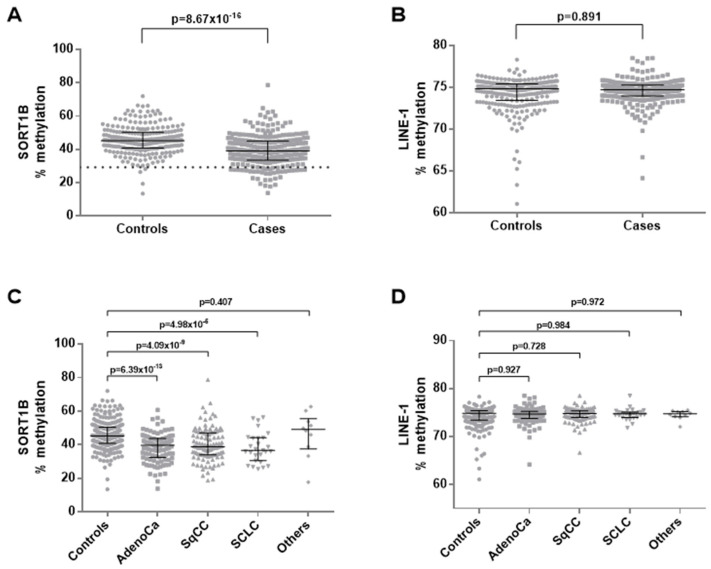
DNA methylation analysis of PBMCs from lung cancer cases and matched controls. DNA methylation of SORT1B promoter demonstrates a low (6% average) but very consistent reduction in the cases (**A**), while global DNA methylation levels are not different (**B**,**D**). Interestingly, SORT1B promoter methylation is low across all three major lung cancer histological subtypes (**C**).

**Figure 5 cancers-16-02154-f005:**
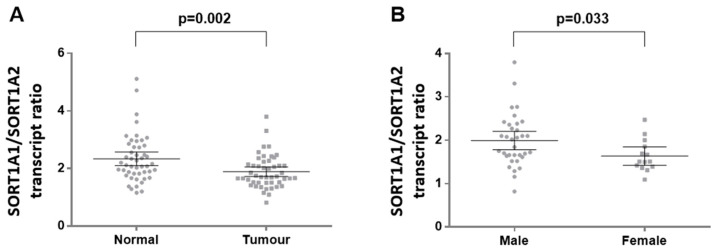
SORT1A exon-8 splice variant expression in primary NSCLC and paired adjacent normal lung tissue. The ratio SORT1A1:SORT1A2 is significantly decreased in lung tumours compared to normal lung tissue (**A**). Clinicopathological parameters demonstrate a borderline association of SORT1A1:SORT1A2 ratio is lower in females compared to males (**B**).

## Data Availability

Data are available to share under an appropriate DTA.

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
