# Peer review of "Novel Transcriptional and DNA Methylation Abnormalities of SORT1 Gene in Non-Small Cell Lung Cancer"

_cancers, 2024, doi:10.3390/cancers16112154_

Round 1
Reviewer 1 Report
Comments and Suggestions for Authors
1) The authors investigated the DNA methylation as an epigenetic mark on the SORT 1 gene. I suggest bringing DNA methylation into the title instead of epigenetic. “ DNA methylation abnormalities”
2) The authors reported only qualitative results in the abstract. E.g. “ Significant hypermethylation of SORT1B promoter was detected in lung tumors” (lines 31-52). I suggest the authors revise the abstract by adding the quantitative results.
3) The authors usually write about the aim of the work in the last paragraph of the introduction. However, in this MS the authors wrote about their methodology and results. (lines 80-85). This should be revised.
4) I strongly recommend the authors draw a graphical abstract at the end of the introduction to transfer the knowledge to the audience.
5) Are the patients who participated in this study just newly diagnosed or are some under treatment also? The authors should describe this. Using mixed patients ( before and under treatment) makes the judgment very difficult since the treatment changes the epigenetic profile.
6) Why did the authors use cell lines? They should be described in section 2.2
7) Why did the authors use scatterplots instead of columns?
8) Some figures like Fig 2 and Fig 4 do not have good resolution. It is required to improve them.
9) The authors claimed ( line 320) that expression balance SORT1B and SORT1A significantly changes in NSCLC. Is it possible to pay attention more to how it changes? Describe more about what is the molecular regulation between SORT1 promoter methylation and NSCLC mortality.
Author Response
Response to Reviewer 1
We thank the reviewer for the constructive comments. We took on board all the possible suggestions as described below.
1) The authors investigated the DNA methylation as an epigenetic mark on the SORT 1 gene. I suggest bringing DNA methylation into the title instead of epigenetic. “ DNA methylation abnormalities”
Title has been revised accordingly.
2) The authors reported only qualitative results in the abstract. E.g. “ Significant hypermethylation of SORT1B promoter was detected in lung tumors” (lines 31-52). I suggest the authors revise the abstract by adding the quantitative results.
Quantitative information added in the abstract
3) The authors usually write about the aim of the work in the last paragraph of the introduction. However, in this MS the authors wrote about their methodology and results. (lines 80-85). This should be revised.
Last paragraph revised accordingly.
4) I strongly recommend the authors draw a graphical abstract at the end of the introduction to transfer the knowledge to the audience.
Graphical abstract provided.
5) Are the patients who participated in this study just newly diagnosed or are some under treatment also? The authors should describe this. Using mixed patients ( before and under treatment) makes the judgment very difficult since the treatment changes the epigenetic profile.
This is indeed very important point. The blood samples from all the patients in this study were received at diagnosis or immediately prior to surgery and prior to any other form of therapy, were from cancer-free controls. Similarly surgical samples were received prior to any chemo or radiation treatment. It is mentioned in line 88 that patients were "chemotherapy naive", but we have now modified the methods section 2.1 accordingly.
6) Why did the authors use cell lines? They should be described in section 2.2
The sentence " SORT1 status was examined in NSCLC cell lines in order to determine if this is an important characteristic for the retention of a malignant phenotype in culture environment." has been added to the beginning of section 2.2.
7) Why did the authors use scatterplots instead of columns?
The scatterplots we used serve as alternative boxplots, with median and interquartile ranges denoted by the lines and all raw values shown. We believe that such plots provide a much better representation of the distribution of the values and allow reader to evaluate better the variance and distribution.
8) Some figures like Fig 2 and Fig 4 do not have good resolution. It is required to improve them.
Figures of better resolution have been provided.
9) The authors claimed (line 320) that expression balance SORT1B and SORT1A significantly changes in NSCLC. Is it possible to pay attention more to how it changes? Describe more about what is the molecular regulation between SORT1 promoter methylation and NSCLC mortality.
The sentence (line 334 in the revised manuscript) has been rephrased to " Our analysis demonstrated that the expression ratio SORT1A/SORT1B significantly increases in NSCLC".
The survival information in our own dataset was limited (as mentioned in line 214); this is why we analysed the expression data in KMPlotter and Xena public domains. These do not contain DNA methylation data to allow the suggested analysis.
A sentence was added (line 246) : " No associations between survival and SORT1B promoter methylation could be attempted as KMplotter and Xena platforms do not hold DNA methylation information".

Reviewer 2 Report
Comments and Suggestions for Authors
The paper investigates the transcriptional and epigenetic aberrations of the SORT1 gene in non-small cell lung cancer (NSCLC), and its potential role in tumor development and progression. The changes in SORT1 expression levels were found to be associated with patient prognosis, where lower expression was correlated with poorer overall and disease-free survival in adenocarcinoma patients.
The key issues that need to be addressed or improved are:
1.Lack of functional validation: While the study revealed the aberrant expression and epigenetic changes of SORT1 in NSCLC, it did not provide direct functional evidence to explain how these changes impact tumor progression or treatment response. Further experiments are needed to elucidate the functional differences of these SORT1 transcript variants and how they regulate key signaling pathways, such as EGFR.
2.Description of control individuals: The description of the "control individuals from the LLP biobank" is too simplistic. The specific composition of this control group needs to be clarified.
3.Formatting issue: The p-value in Line 176 (p=2.26x10-12 Figure 1E) should be presented with proper punctuation.
4.Grammatical error: The sentence in Line 181 ("Interestingly, the correlation between SORT1A and SORT1B transcript levels observed in normal lung (C) is weaker in NSCLC tissue.") has grammatical issues and is not expressed clearly.
Comments on the Quality of English LanguageThe paper investigates the transcriptional and epigenetic aberrations of the SORT1 gene in non-small cell lung cancer (NSCLC), and its potential role in tumor development and progression. The changes in SORT1 expression levels were found to be associated with patient prognosis, where lower expression was correlated with poorer overall and disease-free survival in adenocarcinoma patients.
The key issues that need to be addressed or improved are:
1.Lack of functional validation: While the study revealed the aberrant expression and epigenetic changes of SORT1 in NSCLC, it did not provide direct functional evidence to explain how these changes impact tumor progression or treatment response. Further experiments are needed to elucidate the functional differences of these SORT1 transcript variants and how they regulate key signaling pathways, such as EGFR.
2.Description of control individuals: The description of the "control individuals from the LLP biobank" is too simplistic. The specific composition of this control group needs to be clarified.
3.Formatting issue: The p-value in Line 176 (p=2.26x10-12 Figure 1E) should be presented with proper punctuation.
4.Grammatical error: The sentence in Line 181 ("Interestingly, the correlation between SORT1A and SORT1B transcript levels observed in normal lung (C) is weaker in NSCLC tissue.") has grammatical issues and is not expressed clearly.
Author Response
Response to Reviewer 2
We thank the reviewer for the constructive comments. We took on board all the possible suggestions as described below.
1.Lack of functional validation: While the study revealed the aberrant expression and epigenetic changes of SORT1 in NSCLC, it did not provide direct functional evidence to explain how these changes impact tumor progression or treatment response. Further experiments are needed to elucidate the functional differences of these SORT1 transcript variants and how they regulate key signaling pathways, such as EGFR.
Indeed, our study does not provide functional evidence, as explained in the last paragraph of the discussion. It is an observational study exploring comprehensively the spectrum of abnormalities of SORT1 but not attempting functional associations, which are subject of a subsequent (experimental) type of study, where SORT1 expression will be manipulated. Still, our study presents clear evidence for the extend of SORT1 abnormalities in NSCLC, which, we believe, will trigger those functional studies.
We now emphasise how our studies inform the functional studies required, by adding the sentence “However, it provides important context for such studies and highlights the importance of considering the multiple forms of sortilin.”
2.Description of control individuals: The description of the "control individuals from the LLP biobank" is too simplistic. The specific composition of this control group needs to be clarified.
The phrase " which are recruits from the population cohort element of the LLP study that have confirmed absence of lung cancer for at least 5 years after the blood sample was taken" was added to further describe the controls used.
3.Formatting issue: The p-value in Line 176 (p=2.26x10-12 Figure 1E) should be presented with proper punctuation.
Thanks for spotting this. We corrected it along with a few other instances in the manuscript where superscript formatting was lost. (line 189 and others in the revised manuscript).
4.Grammatical error: The sentence in Line 181 ("Interestingly, the correlation between SORT1A and SORT1B transcript levels observed in normal lung (C) is weaker in NSCLC tissue.") has grammatical issues and is not expressed clearly.
We rephrased the sentence to: " Interestingly, the correlation coefficient between SORT1A and SORT1B transcript levels in normal lung (C) is higher than the one observed in in NSCLC tissue (D), indicating a potential distortion of SORT1A/1B transcription in cancers." - line 197 of revised manuscript.
Reviewer 3 Report
Comments and Suggestions for Authors
The manuscript focused on Novel transcriptional and epigenetic abnormalities of SORT1 2 gene in non-small cell lung cancer represents a technically correct and timely relevant manuscript available for the publication on this journal after minor considerations
- In the introduction section, please, could the authors display the opening challenges for NSCLC patient's clinical stratification?
- In the methodological section, please, could the authors add other histological details about NSCLC samples?
- In the text, please, could the authors consider the role of entire molecular pathway involving SORT1? It could be interesting evaluate the signaling pathway modifications according to SORT1 epigenetic changes.
- Please, could the authors report in italics gene name?
Comments on the Quality of English LanguageMinor english editing
Author Response
Response to Reviewer 3
We thank the reviewer for the constructive comments. We took on board all the possible suggestions as described below.
- In the introduction section, please, could the authors display the opening challenges for NSCLC patient's clinical stratification?
We have added the following sentence at the end of paragraph 1 in the introduction:
" Some molecular abnormalities established in NSCLC are important determinants for clinical stratification for targeted therapies. However, the major challenge is heterogeneity and the multiplicity of those molecular abnormalities, which essentially create very diverse clinical phenotypes. Identification of common abnormalities provides additional opportunities for new treatments, improved stratification, or earlier diagnosis.".
- In the methodological section, please, could the authors add other histological details about NSCLC samples?
Thanks for spotting this; we added a sentence to the section, highlighting where these details can be found and providing a brief overview): Clinicopathological characteristics of the patients used for expression and methylation studies are provided in Supplementary Table 1
- In the text, please, could the authors consider the role of entire molecular pathway involving SORT1? It could be interesting evaluate the signalling pathway modifications according to SORT1 epigenetic changes.
Thank you for this interesting question. One could postulate that these changes may alter sortilin’s interaction with other co-receptors such as p75NTR and Trk receptors and impact on the signalling pathways associated with these receptors. We have included a paragraph in the discussion (2nd paragraph).
Round 2
Reviewer 1 Report
Comments and Suggestions for Authors
The authors answered my comments and the quality of the manuscript has been improved. However, they must provide a graphical abstract at the end of the introduction.